# Soil Moisture Content from GNSS Reflectometry Using Dielectric Permittivity from Fresnel Reflection Coefficients

**Andres Calabia [1,2] , Iñigo Molina [1,2,* ] and Shuanggen Jin [1,3]**

1   School of Remote Sensing and Geomatics Engineering, Nanjing University of Information Science and Technology, Nanjing 210044, China, andres@calabia.com (A.C.); sgjin@nuist.edu.cn (S.J.)
2   School of Land Surveying, Geodesy and Mapping Engineering, Universidad Politécnica de Madrid, South Campus, 28031 Madrid, Spain
3   Shanghai Astronomical Observatory, Chinese Academy of Sciences, Shanghai 200030, China
*   Correspondence: inigo.molina@upm.es

**Abstract:** Global Navigation Satellite Systems-Reflectometry (GNSS-R) has shown unprecedented advantages to sense Soil Moisture Content (SMC) with high spatial and temporal coverage, low cost, and under all-weather conditions. However, implementing an appropriated physical basis to estimate SMC from GNSS-R is still a challenge, while previous solutions were only based on direct comparisons, statistical regressions, or time-series analyses between GNSS-R observables and external SMC products. In this paper, we attempt to retrieve SMC from GNSS-R by estimating the dielectric permittivity from Fresnel reflection coefficients. We employ Cyclone GNSS (CYGNSS) data and effectively account for the effects of bare soil roughness (BSR) and vegetation optical depth by employing ICESat-2 (Ice, Cloud, and land Elevation Satellites 2) and/or SMAP (Soil Moisture Active Passive) products. The tests carried out with ICESat-2 BSR data have shown the high sensitivity in SMC retrieval to high BSR values, due to the high sensitivity of ICESat-2 to land surface microrelief. Our GNSS-R SMC estimates are validated by SMAP SMC products and the results provide an R-square of 0.6, Root Mean Squared Error (RMSE) of 0.05, and a zero p-value, for the 4568 test points evaluated at the eastern region of China during April 2019. The achieved results demonstrate the optimal capability and potential of this new method for converting reflectivity measurements from GNSS-R into Land Surface SMC estimates.

**Keywords:** Soil Moisture Content (SMC); Global Navigation Satellite Systems Reflectometry (GNSS-R); CYGNSS; Soil Moisture Active Passive (SMAP); Fresnel reflection coefficients

## 1. Introduction

In many different scientific fields, the utmost significance of Soil Moisture Content (SMC) has been pointed out as an environmental factor for land surface dynamics monitoring [1–8], such as evapotranspiration, droughts, floods, etc., while it simultaneously regulates energy and water exchange between the land and the atmosphere and other hydrological processes. SMC also enables the regulation of infiltration rate due to precipitation and thus controlling water runoff and surface erosion. Over a large scale, SMC is a key parameter for climate change studies [9]. In addition, SMC coupled with other environmental variables, such as land surface temperature, land cover, and precipitation, among others, are commonly used as input of climate models [10–12]. Moreover, in agriculture, SMC is a crucial indicator of plant growth and crop yield, as well as for drought prediction. Therefore, since all these variables are strongly linked to social and economic impacts on human activity [11], SMC is being employed for planning and managing irrigation strategies [4].

SMC can be estimated from a wide variety of methods and techniques. For instance, for local observations, direct procedures include, e.g., TDR (Time Domain Reflectometry) electric permittivity [2]. However, these local methods are only suitable for small areas (~km$^2$) and the time needed for surveying is unacceptable for practical application on regional and meso-scales.

Fortunately, in these last decades, near-Earth satellites have provided an unprecedented opportunity to sense SMC from space and with a wide diversity of techniques and sensors (e.g., [5,13]). Although remote sensing observing activities began in the 1970s, technological development has evolved exponentially to get benefit from advances on electromagnetic (EE.MM.) radiation through different spectrum intervals [11]. Two main spectral domains for assessment of SMC can be devised, usually referred to optical and microwave spectrum, each of them with particular advantages and limitations [5,14,15]. In this study, we work in the microwave domain, where the wavelength varies from some few centimeters to meters, commonly employed in the ranges of λ = 2.5–3.75 cm (band X at 8.0–12 GHz), λ = 3.75–7.5 cm (band C at 4–8 GHz), and λ = 15–30 cm (band L at 1–2 GHz). In this domain, measurements can either be passive or active. With the passive instruments, soil dielectric properties and soil brightness temperatures can be retrieved for SMC assessment. With the active devices, only soil dielectric properties can be assessed by the means of measuring the scattering coefficient, usually denoted as the bistatic or back-scattering coefficient [16,17]. Namely, a bistatic system holds when the transmitting and the receiving antennas are separately placed or included as independent instruments. In a mono-static system, the antenna alternates between transmitting and receiving signals, so only one instrument is required. Most of Earth observing systems belongs to the mono-static configuration. Unfortunately, SMC cannot be directly retrieved from active or passive microwave remote sensors, and different physical models based on geometrical and/or physical and optical properties have to be built [16,17]. Active or passive signals have the advantage of moderately penetrating soil surfaces when they are not severely limited by atmospheric conditions. However, in both cases, the signals are strongly conditioned by geophysical and biophysical variables such as Bare Soil Roughness (BSR) and the different constituents of the vegetation canopy [18]. A complete description of active and passive systems for SMC estimation, including frequency bands and spatial resolutions, is given in [13].

In the same frequency domain, during the last few decades, an emerging and challenging technology based on the opportunity signal, i.e., receiver devices that take the advantage of employing existing signals from other systems, is being exploited for specific Earth observation applications such as, e.g., geoid determination, sea surface wind speed, and surface SMC [19]. Among these, the technology which employs Global Navigation Satellite Systems (GNSS) is referred as GNSS Reflectometry (GNSS-R). Current GNSS include the constellations of the United States (GPS NAVSTAR), Europe (Galileo), Russia (GLONASS), and China (BeiDou). These systems were originally designed for navigation, positioning and synchronization of time [20]. GNSS-R is based on measurements reflected by GNSS signals on the Earth's surface, which are employed to estimate geophysical parameters [21–23]. Although the current GNSS-R missions were initially designed and conceived to observe ocean altitudes, winds, and tropical cyclones, recent studies have exhibited the capabilities for sensing land surface attributes [24–32]. The first missions carrying GNSS-R instruments started in 2003, with the UK Disaster Monitoring Constellation (UK-DMC) mission [33]. Currently, this technology is developing rapidly with the launch of TechDemoSat-1 in 2014, the current NASA's (National Aeronautics and Space Administration) Cyclone GNSS (CYGNSS) since 2012, the 3CAT-2 of Polytechnic University of Catalonia (UPC), and the GEROS (GNSS REflectometry, Radio Occultation, and Scatterometry) experiment of the European Space Agency (ESA), among others [34]. It is clear that GNSS-R arouses a growing interest among the scientific community.

A GNSS-R system holds the same geometrical configuration properties of a bistatic radar [35]. The principle of measurement is based on observing with a GNSS-R receiver the quasi-specular Left-Hand Circularly Polarized (LHCP) reflection of a Right-Hand Circularly Polarized (RHCP) GNSS signal [31,36]. In this case, a perfectly smooth surface will produce a near specular reflection,

while a rough surface will spread the transmitted signal over a larger area producing a scattered signal. The scattered signal is sampled over the illuminated zone in delay and frequency, creating the so-called Delay Doppler Map (DDM), which is the basic product containing physical information of a surface [34,36]. One important application of GNSS-R is the measurement of surface reflectivity, which allows the assessment of soil dielectric properties and to derive SMC [35,37]. An important property of the returned GNSS signals is the coherency parameter, being high for nearly smooth surfaces, and low for very rough surfaces. This parameter, namely called BSR, has an important implication for the modeling of signals when geophysical land surface properties need to be estimated [31,37]. In addition to BSR, when retrieving SMC over vegetated land, an additional parameter related to the specific vegetation canopy must be taken into account [38]. Therefore, since current GNSS-R missions does not provide BSR nor vegetation canopy characteristics yet, this information need to be accessed from other existing sources, such as NASA's Ice, Cloud, and land Elevation Satellites 2 (ICESat-2) [39], providing BSR estimates, or NASA's Soil Moisture Active Passive (SMAP) [40], providing BSR and vegetation biophysical variables such as Vegetation Optical Depth (VOD) or Vegetation Water Content (VWC).

According to previous work, and taking into account the aforementioned analyses, in this paper, our study is motivated by the potential capability of CYGNSS for estimating SMC. In this field, studies have mainly focused their efforts in deriving and comparing CYGNSS reflectivity values with external sources of SMC such as SMAP SMC estimates, either by direct comparison, statistical regressions, or time series analyses [19,27,37,41–44]. In this work, we go one step ahead on employing the CYGNSS coherent component for deriving reflectivity estimates, and correct from BSR and VOD effects. Then, we employ the converted Fresnel linear reflectivity coefficients from GYGNSS reflectivity estimates to derive SMC. This is possible for low incidence angles as it has been pointed out by [31,45]. Under these conditions, our results show the potential capability of the inversion procedure and highlight the conditions for a correct assessment of SMC. In order to validate this approach, specific geographical areas are tested by means of existing SMAP SMC datasets.

This manuscript is organized as follows: In Section 2, we firstly describe the different datasets involved in the SMC retrieving process and the geographical extent for the validation test. Then, our proposed reflectivity conversion method of CYGNSS DDMs to SMC is shown. In Section 3, the experiments and results are presented, and Sections 4 and 5 show the discussion, and conclusions and recommendations for future research.

## 2. Data and Methods

### 2.1. CYGNSS Data

The CYGNSS mission of NASA is the most recent Earth-observing GNSS-R constellation. It consists of eight satellites launched onboard a Pegasus XL Rocket on 15 December 2016 (online measurements are available from March 2017 at https://podaac-opendap.jpl.nasa.gov). The CYGNSS satellites are located at an equatorial orbit with an inclination of 35° at about 500 km altitude. The initial research objectives of CYGNSS were the observation of Tropical Cyclones, so the coverage was limited to low and middle latitude ranges (38°S to 38°N), and consequently restricting the areas of study for other applications. For instance, the surface scattered signals have provided an unprecedented opportunity to study BSR and dielectric properties [36,41,46].

In CYGNSS, the geometry of observation follows a bistatic configuration [47]. Each CYGNSS satellite is equipped with a GNSS up-looking RHCP antenna and two GNSS down-looking LHCP antennas. The antennas correspond to the bistatic radar receivers kind on the L-band (L1-band frequency at 1.57542 GHz), also called DDM instruments. These instruments are designed to map the scattered signal on the ocean and land surface, which is sampled in time and frequency, thus delivering DDMs at the proximity of the specular point (SP). The DDM instrument computes 4 measurements every second, which are compressed and downloaded to the ground processing facilities [46]. There are three available CYGNSS processing levels delivered freely to the scientific community, ranging from

level 0 to 3, and subsequent sub-levels. In this study, we employ level-1 data, which contains geo-located DDMs calibrated into an ideal (analog) power sensor ($P_{DDM}$). Additional level-1 parameters required in this study include DDM timestamp ($t_{UTC}$), latitude ($\varphi_{SP}$), longitude ($\lambda_{SP}$), and incidence angle ($\theta_{SP}$) of the SP, antenna gain ($G_r$), GPS effective isotropic radiated power (eirp), and ranges from the receiver ($R_r$) and transmitter ($R_t$) antennas.

The CYGNSS DDM data specifications and processing schemes are given in [48–50]. The geometric distribution of the observed SPs follows a quasi-random spatiotemporal distribution, due to the changing geometry of GNSS and CYGNSS satellites [41]. This distribution is very different from conventional data delivered by active/passive microwave instruments onboard satellites. Therefore, the challenge includes very different conditions of the first Fresnel ellipses (SP areas), because the distances from the transmitter and the receiver differ considerably from one observation to others. In addition, these areas may vary according to other factors, including the incidence angle and the height of the receiver [47,51,52]. The authors in [46] quantified the corresponding mapping resolution near to 150 m depending on the observation geometry. The authors in [53] assessed the spatial resolution of the coherent Fresnel reflection zone to approximately $0.65 \times 0.85$ km$^2$, also depending on the above factors. The rationale in this manuscript is to derive CYGNSS reflectivity values and combine them with other necessary parameters for SMC estimation. In this case, we employ estimates of VWC and BSR from NASA's SMAP and ICESat-2 missions. Figure 1 shows an example of CYGNSS' SPs distribution from the eight CYGNSS satellites on 1 April 2019, together with ICESat-2 data. ICESat-2 data is introduced in the next section.

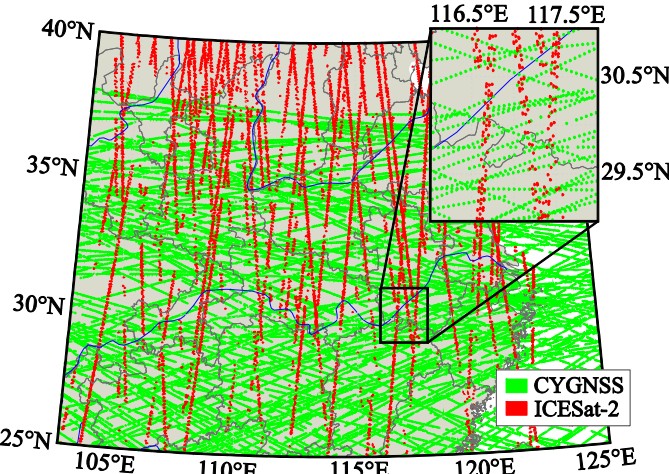

**Figure 1.** The locations of CYGNSS's SPs from its eight satellites are plotted in green for 1 April 2019. ICESat-2 data from the whole month of April 2019 is plotted in red. Zoom window shows the spatial resolution in more detail.

*2.2. ICESAT and SMAP Data*

In order to retrieve SMC from CYGNSS observables, ancillary or reference BSR and VWC data must be taken into account, so their respective attenuations are compensated in the recorded signal. The reflectivity values that can be estimated by CYGNSS do not account for these contributions, and they must be corrected from VWC. The ICESat-2 and the SMAP missions can provide these parameters, respectively. Then, we can also employ SMAP's SMC estimates to validate the methodology of GNSS-R SMC retrieval presented in this manuscript. Note that SMAP SMC estimates have been computed using SMAP BSR and VWC products and computing CYGNSS SMC estimates with other products than SMAP BSR and VWC might provide discrepancies in the validation. Therefore, we assume that employing SMAP BSR and VWC for CYGNSS SMC validation is the preferred scheme. Notwithstanding, an equivalent estimate of VWC could be derived from other sources, e.g., Landsat [30,54,55], making our SMC retrieval completely independent from the SMAP mission, and with possible improvements.

In addition, more accurate SMC products could be obtained with the use of, for example, the high accuracy ICESat-2 BSR estimates.

BSR is a key variable for surface scattering models [56–59]. Generally, BSR is estimated in-situ by more or less sophisticated instruments that allow measuring, at the best, 4 m length 2-dimensonal BSR profiles. These methods are incapable of measuring a large surface extent. In addition, BSR can also be estimated by computing accurate 3-dimensional surface models. These traditional procedures usually measure short local profiles and are not suitable for large areas, as measured by Low Earth Orbit (LEO) GNSS-R satellites, due to incompatible spatial resolution. Besides, vegetated areas, such as agricultural crops, meadows, shrub areas, etc. are not accessible by ground-based techniques. Moreover, in-situ measurements might be unnecessary since GNSS-R deals with L bands, so BSR below 19 cm is imperceptible [35]. In this study, we include a revision of this hypothesis by investigating to what extent BSR must be considered for L band SMC retrieval. Two alternatives are suggested, the SP area and the system frequency band.

Since October 2018, the second generation of NASA's ICESat-2 has been operating globally, supplying Earth's surface geophysical and vegetation parameters. ICESat-2 data is acquired by the Advanced Topographic Laser Altimeter System (ATLAS), which operates in the 532 nm band, with a multi-beam micro-pulse laser, also referred to as photon-counting, from which the surface observable range is derived from the travel time of each detected photon. Helped by GNSS, each photon can be accurately located on Earth's surface [39]. In addition to surface height measurements, ICESat-2 supplies BSR and canopy parameters, very valuable data for microwave scattering research [39]. ICESat-2 measurements are processed every 100 m data segments, containing more than 100 signals of photons. Thus, according to this high-density sampling, the fulfillment of the required spatial resolution CYGNSS SMC retrieval might be achieved. ICESat-2 products are available at https://nsidc.org/data/atl08. However, due to possible BSR changes throughout the year, we only employ data from April 2019. Figure 2b shows a subset of the ICESat-2 data employed in this study.

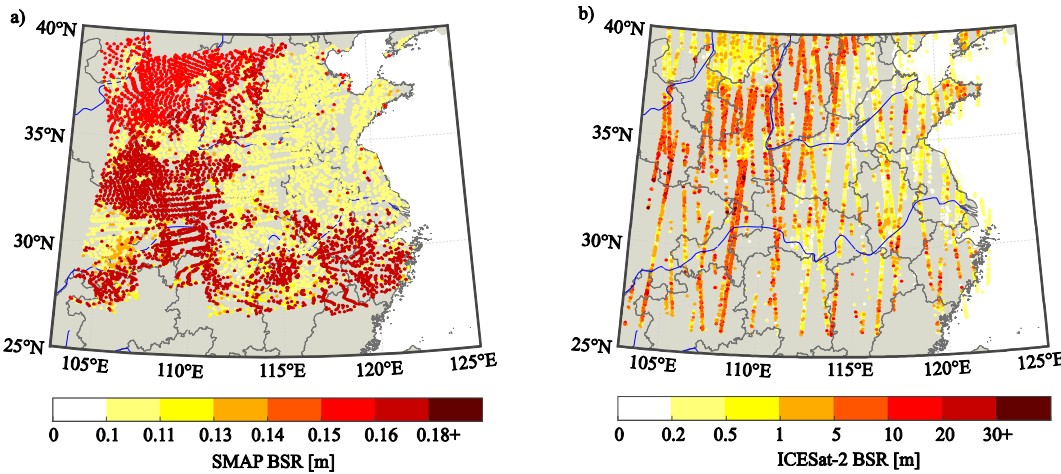

**Figure 2.** In (**a**), SMAP BSR estimates for the period 1–13 of April 2019 plotted at 500-sample intervals. In (**b**), ICESat-2 BSR estimates for the whole month of April 2019 plotted at 100-sample intervals.

The SMAP/Sentinel-1 L2 Radiometer/Radar 30-Second Scene 3 km EASE-Grid Soil Moisture (Version 2) product [40] provides SMC estimates for the first 5 cm of surface depth. In addition, as mentioned above, it also provides VWC with a ±0.04 volumetric accuracy (VWC < 5 kg m$^{-2}$). Other relevant reference information found in the SMAP products are VOD, also key parameter for microwave scattering model assessment and attenuation of related effects on reflectivity. In addition, we employ SMAP SMC estimates as reference information for validating our GNSS-R SMC retrieval technique (see next section). SMAP products are available at https://nsidc.org/data/SPL2SMAP_S/versions/2. Figures 2a and 3 show the SMAP BSR and VOD estimates, respectively, for the period 1–13 of April

2019. Finally, the locations of SMAP and ICESat-2 estimates are compared to CYGNSS SP locations, and we select the SMAP and ICESat-2 estimates located in a range inferior of 500 m from each CYGNSS SP location.

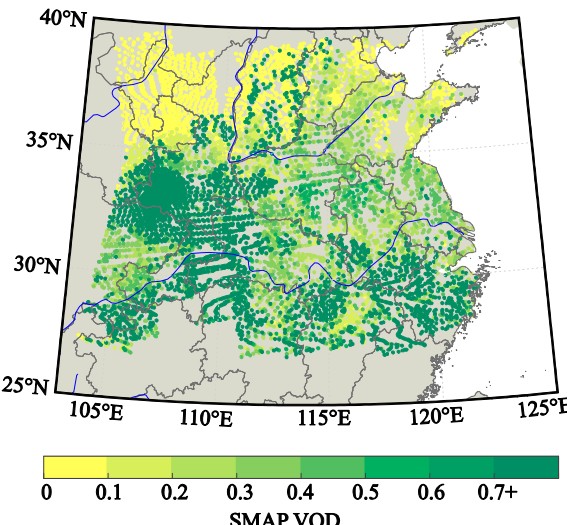

**Figure 3.** SMAP VOD estimates at 500-sample interval for the period 1–13 of April 2019. Units are dimensionless.

## 2.3. Study Area and Validation Scheme

The study area covers a large region across the Central and Eastern parts of the Popular Republic of China. Figure 4 shows the land cover map (National Institute for Environmental Studies, Tsukuba, Japan, http://db.cger.nies.go.jp/dataset/landuse/en/) [60] of the study area where the main land-cover types include, from West to East, bare soil, grassland, and agriculture in the North, and agricultural and woodland, in the South.

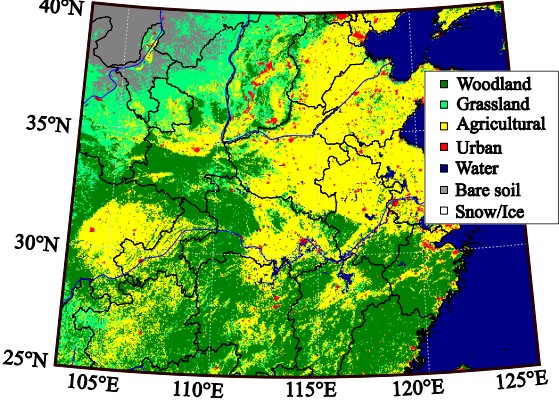

**Figure 4.** Landcover map of the study area (available at the National Institute for Environmental Studies, Tsukuba, Japan, http://db.cger.nies.go.jp/dataset/landuse/en/).

The study area is approximately bounded between latitudes 26°N and 40°N, and longitudes 103°W and 125°W. The total extension is approximately $3.6 \times 10^6$ km$^2$ ($1500 \times 2400$ km$^2$). Due to its large extension, several continental climates are affecting the different regions. For instance, in the North-West region, e.g., Mongolia Plateau, or in the East region, e.g., influenced by the ocean. In general, two climates can be differentiated in our study area, humid subtropical without dry season (Cfa) in the South, and humid subtropical with summer monsoon and dry winter (Cwa) in the North. Humid continental climates at North-East and semi-arid climates at North-West also can be found in the most Northern regions of the study area. However, for SMC assessment and the scope of this

study, these climatic differences are negligible, since we focus on the measurement of the variable itself, including the vegetation influences.

In Figure 4, natural and agricultural areas can be perfectly identified. For instance, the main agricultural areas can be differentiated in the North-East region, and more naturally vegetated woodland in the South. The main crops cultivated in the agricultural areas are corn and other types of cereals with a typical phenological period, i.e., winter crops, rice in the South and wheat and corn in the northern regions of the study area. Note that during the month of April, these crops are in the growing stage, due to the precipitation regime. This situation is particularity interesting for quantifying the effects of the different mentioned variables on the CYGNSS signal subject to analysis.

In this study, we employ SMAP SMC estimates to validate our GNSS-R SMC retrieval technique. Figure 5 shows the spatial distribution of SMAP SMC estimates for April 2019, where different subareas can be identified from the day-to-day data collection. Since SMC is sensible even for day-to-night variations and possible precipitations, we classify the data analysis by days and limit our SMC validation at a maximum time difference between SMAP and CYGNSS times of acquisition of about 3 h. We assume that during this ±3 h range the SMC state may be considered nearly stable under the absence of precipitation. The information regarding precipitations has been accessed through the China Meteorological Data Service Center (https://data.cma.cn/eng), which delivers global precipitation regime maps for the provinces of the People's Republic of China in a 24 h basis, among other options. Finally, in order to select SMAP estimates coincident with CYGNSS SP locations for the validation test, we select the SMAP estimates located in a range inferior of 500 m from each CYGNSS SP location.

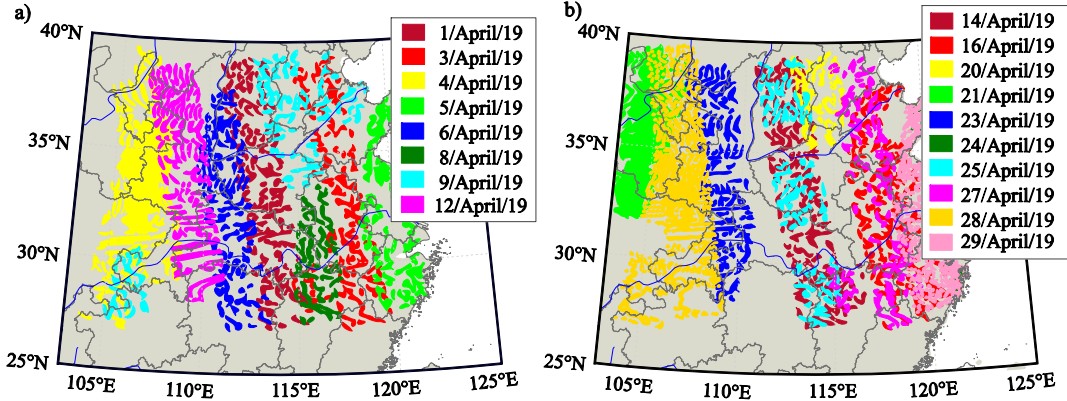

**Figure 5.** SMAP data for the periods (**a**) 1–12 and (**b**) 14–29 of April 2019 plotted at intervals of 500 samples.

## 3. Methodology

In recent years, different approaches have been addressed for GNSS-R SMC retrieval [26,32,33,38, 46]. These approaches usually employ a modification of the bistatic radar model described in [61,62]; its basic equation can be expressed as [25,57]:

$$P^r_{pq} = P^c_{pq} + P^i_{pq} \tag{1}$$

In this expression, $P^r_{pq}$ is the scattered power received at the reflectometer, and $P^c_{pq}$ and $P^i_{pq}$ are the received coherent and incoherent power, respectively. The subscripts $p$ and $q$ stand for the incident and scattered polarizations, respectively. Regarding this, [46,63] considered that for certain smooth surfaces under specific conditions, the strongest scattering is received from the coherent contribution, relying on the assumption that scattering decreases quickly as it moves away from the nominal SP location. Natural land surfaces are included in this hypothesis and this is the rationale of working with the coherent contribution of GNSS-R.

We employ CYGNSS level 1 data, which was initially designed to study wind and ice over the oceans. However, the above literature has shown its potential to derive land surface reflectivity, which in turn is tightly coupled with SMC (e.g., [19,34,42,44]). In this scheme, for the purpose of this work, we employ GNSS-R models described in [31,42,43,47] for the modeling of CYGNSS DDM reflectivity estimates. Under this assumption, the bistatic radar equation for the coherent component of smooth surfaces from LHCP GPS bistatic microwave signals observed by a receiver can be written as:

$$P_{lr}^c = \frac{P_t \lambda^2 G_t G_r}{(4\pi)^2 (R_t + R_t)^2} \Gamma_l(\theta) \tag{2}$$

In this equation, $P_{lr}^c$ is the DDM peak value from the analog scattered power (namely, $P_{DDM}$). The subscript *rl* stands for a scattering mechanism when the incident RHCP signal is scattered by the surface and inverts the polarization to LHCP at the receiver position. $\Gamma_{lr}$ is the surface reflectivity from which the SMC might be estimated. Solving for reflectivity, Equation (2) can be expressed in the manner specified by [37], which appears more convenient for processing GYGNSS data:

$$\Gamma_{lr}(\theta) = \frac{(4\pi)^2 (P_{DDM} - N)(R_t + R_t)^2}{\lambda^2 G_t G_r P_t} \tag{3}$$

This is the final expression for a CYGNSS SP reflectivity, where measurements of $\Gamma_{lr}$ need to be calibrated from instrumental bias [37]. For the CYGNSS data employed in this study, we employ a reflectivity constant bias value of 0.15. In Equation (3), we introduced a new variable, $N$, which refers to the power noise floor of the DDM. For any surface, in general, but in particular for land surfaces, this observed reflectivity is affected by its corresponding BSR parameter. The implementation of the BSR effect in Equation (3) is:

$$\Gamma_{lr}(\theta) = \left| R_{lr}(\theta) \right|^2 \chi(z) \tag{4}$$

In this equation, $R_{lr}(\theta)$ is the Fresnel reflection coefficient and $\chi(z)$ is the probability function for the surface height z and can be modeled by the Gaussian probability density function [47].

Moreover, when natural surfaces are covered with vegetation, the observed reflectivity is attenuated. Therefore, this contribution [38] must be taken into account in Equation (4) as:

$$\Gamma_{lr} = \left| R_{lr}(\theta_i) \right|^2 e^{(-2k\sigma \cos \theta_i)^2} e^{-2\tau/(\cos \theta_i)^2} \tag{5}$$

In this equation, $\tau$ stands for the VOD and $h = 4k^2\sigma^2$, with $k = 2\pi/\lambda$, where $\lambda$ is the wavelength of the system, and $\sigma$ is the standard deviation of the surface [64]. This is the expression for surface reflectivity corrected by BSR and VOD and shows that both effects of soil and vegetation are relevant for SMC determination. As mentioned above, in this study we employ and compare CYGNSS SMC retrieval using both ICESat-2 and SMAP BSR estimates.

Regarding the VOD parameter $\tau$, the main factors influencing this variable are the signal polarization and the incidence angle ($\theta_i$) [46]. Note that, since for low vegetated areas the VOD parameter $\tau$ holds a linear relation with the VWC, it can be related to the Normalized Difference Vegetation Index (NDVI), or other spectral indices [30,54,55]. Despite many efforts that have been done for evaluating VWC by means of passive microwave data [11,65,66], the VOD parameter $\tau$ remains difficult to estimate. This is due to its linear relation to VWC, by means of a scaling parameter, which is function of the type and structure of the canopy, and the polarization and wavelength of the signal, which is estimated experimentally [65,67]. Unfortunately, the problem of determining VOD is to find a proper relation with VWC. The authors in [65] revealed that, independently of the kind of vegetation, only a small variation of a certain scale factor produces an insignificant error for VOD in the L band range (18–21 cm). In any case, VOD is correlated to VWC under experimental circumstances, which is the case in SMAP/Sentinel 1 L2 data, and the use of this variable is preferred. Further studies on this

hypothesis can be found in [35,46], showing that leaves of vegetation are almost transparent for the L band, and attenuation is mainly due to other vegetation constituents, such as branches or trunks.

Then, the reflectivity corrected from BSR and VOD (equivalent to VWC), i.e., $R_{lr}(\theta)$, is employed to estimate soil permittivity ($\varepsilon_r$) by means of the Fresnel reflection coefficients, that are related to the reflectivity linear polarization modes [31,45,47,51]:

$$R_{lr} = R_{rl} = \frac{1}{2}(R_{vv} - R_{hh}) \tag{6a}$$

$$R_{rr} = R_{ll} = \frac{1}{2}(R_{vv} + R_{hh}) \tag{6b}$$

In these equations, $R_{vv}$, and $R_{hh}$ are the Fresnel coefficients for vertical and horizontal polarization, respectively. The subscripts, lr and *rl* hold for circular cross-polarized reflections, while *rr* and *ll* for co-polarized reflections. For soil surfaces, the Fresnel reflectivity ($R_{vv}$ and $R_{hh}$) is function of soil permittivity or dielectric constant $\varepsilon_r$ and the incidence angle ($\theta$), from which SMC is retrieved, i.e., $SMC = f(\varepsilon_r)$. The conversion from the Fresnel reflectivity to the real part of permittivity ($\varepsilon_r$) has no direct solution. However, [31] suggests two solutions that allow solving either for $R_{vv}$ or $R_{hh}$, based on the models given in [45] or [67], respectively. These models are suitable for low incidence or high elevation angles, which is the case of satellite viewing properties and, particularly, for observations of GYGNSS SPs. Once the dielectric constant is retrieved, the volumetric SMC can be derived from the Topp model [68,69]. The advantage of this model is that clay, sand, and silt proportions, i.e., textural properties of the soil, are unnecessary.

In summary, the SMC retrieval methodology suggested in this study is presented in Figure 6 as follows:

1. Values for Equation (3) can be obtained from GYGNSS products.
2. BSR values are obtained from ICESTA2 or SMAP products.
3. VWC or VOD are obtained from the SMAP products.
4. Surface reflectivity $R_{lr}(\theta)$ is corrected by means of the previous values using Equation (5).
5. $R_{vv}$ and $R_{hh}$ Fresnel coefficients (Equation (6)) are solved for low incidence angles ($\theta_i < 35°$) where $|R_{vv}| = |R_{hh}|$.
6. The real part of permittivity ($\varepsilon_r$) is solved by either the method of [45] for $R_{vv}$ or by the method of [67] for $R_{hh}$.
7. SMC is finally derived applying the Toppmodel [68].

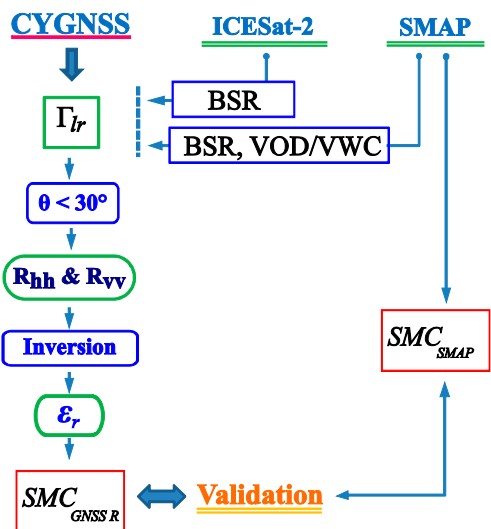

**Figure 6.** Flowchart of GNSS-R SMC retrieval methodology.

## 4. Results

In this manuscript, we have derived SMC estimates from Fresnel reflection coefficients measured by GNSS-R, specifically, from the CYGNSS mission during April 2019 for the eastern region of China. In summary, the conversion of GNSS-R data into SMC values has been carried out according to the methodology specified in Section 3 and summarized in Figure 6. In brief, first, CYGNSS reflectivity measurements ($\Gamma_{lr}$) have been corrected from BSR and VOD (equivalent to VWC) effects by implementing a probability function into the bistatic radar model. Afterwards, the Fresnel reflectivity ($R_{hh}$) has been converted into dielectric permittivity ($\varepsilon_r$) and, subsequently, into volumetric SMC. Two different approaches are employed for the SMC estimation, one with BSR estimates from ICESat-2 and the other with BSR estimates from SMAP. While ICESat-2 BSR estimates appear to have a better spatial resolution, their measurement footprints are sparsely distributed on both sides along the satellite ground tracks. In turn, it results in large data gaps between orbit legs that can only be resolved by employing the precession of the satellite orbit during a large enough period of time (see Figure 2b). Despite the ICESat-2 lack of coverage, an attempt has been made to verify the suitability of the BSR data provided by such a system for its application in the bistatic SMC retrieval model, which will also help to understand the behavior of SMAP BSR estimates. Subsequently, in order to evaluate the suitability of ICESTA2 BSR estimates, we perform a sensitivity analysis of the surface reflectivity (Equation (4)) to this variable. In the following subsections, we present our analysis and validation test results for the suggested GNSS-R SMC retrieval.

### 4.1. SMC Sensitivity Analysis to GNSS-R Reflectivity Γlr, Incidence Angle θ, BSR, and VOD Input Parameters

Figure 7 shows the SMC model sensitivity analysis to incidence angles ($\theta$) of 5°, 15°, 25°, and 35° under different BSR and VOD conditions, for uncorrected GNSS-R reflectivity measurements ($\Gamma_{lr}$) from 0.1 to 0.5 at 0.1 intervals. From this figure, in the top panels (Figure 7a,b), we can observe that SMC values are strongly dependent on a VOD increase from (a) 0.1 to (b) 0.3. In addition, under these conditions and for a similar SMC state, the dependence on incidence angle variations is more pronounced for the lower VOD case (Figure 7a). This can also be observed in the lower panels of Figure 7, whereas SMC converges for higher VOD values. Moreover, in Figure 7a, the incidence angle shows to be a key parameter for SMC estimation at SPs with high reflectivity ($\Gamma_{lr}$). On the other hand, at the lower panels (Figure 7b,c), SMC values show a similar response to BSR variations from 0.1 m to 0.4 m for VOD cases between 0 and 0.3, suggesting the lack of SMC dependence to BSR values below 0.4.

From Figures 7 and 8, we can observe that for a given VOD coverage, surface reflectivity and SMC estimates are not sensitive to BSR values below 0.2 m. These results imply that small variations in surface reflectivity will not influence the SMC retrieval below BSR values of ~0.2 m. On the contrary, Figure 8 clearly shows that GNSS-R reflectivity appears to be more sensitive to BSR values above 0.2 m, which supports the findings of [70] for this variable. Since the maximum value of SMAP BSR for the data sets of this study is not higher than 0.18 m, we can foresee that SMAP BSR values will have a small influence on SMC retrieval, as the permittivity is directly related to the surface reflectivity. However, from Figure 2b, we can observe that for ICESat-2 data, BSR values exhibit a higher variability within the same SMAP areas, which would imply better surface reflectivity estimation. Unfortunately, ICESat-2 values are not available over the complete study area but show the convenience of such types of values for the pursued objective. Nevertheless, an attempt has been made in this work for using ICESat-2 BSR values (Figure 9). Regarding VOD values, a higher sensitivity to this variable is observed when values exceed, which corresponds to some agricultural areas and woodland in Figure 3. This underlines the asseverations of [35] and [46] about agricultural crops and their influence on the L microwave band. Finally, concerning the sensitivity to incidence angle variations, Figure 8 shows that the correction response for an increase of incidence angle under higher BSR values results in a negative reflectivity correction, while under higher VOD values in a positive reflectivity correction.

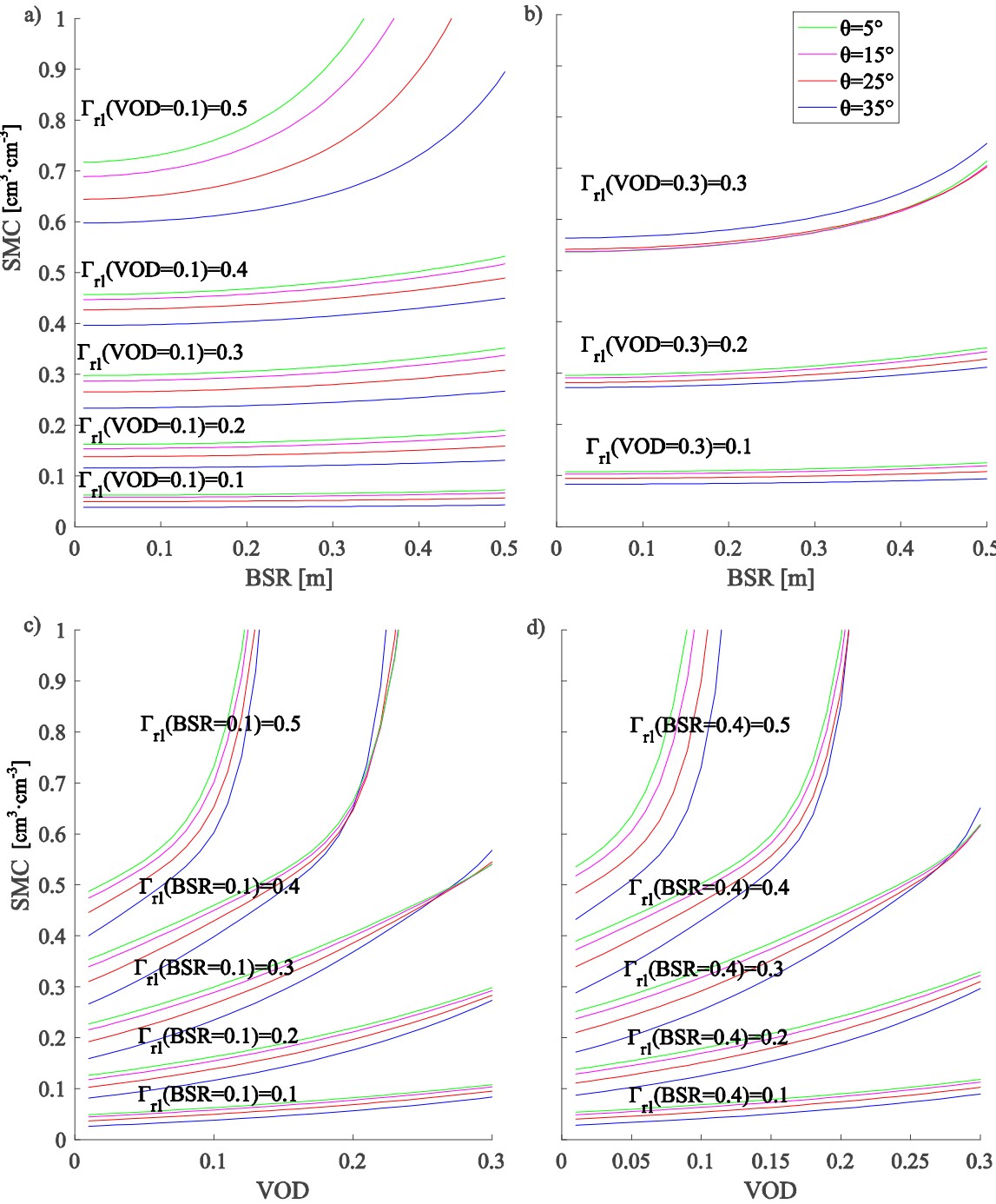

**Figure 7.** SMC reflectivity model sensitivity analysis to variable incidence angle $\theta$, BSR, and VOD for uncorrected reflectivity measurements ($\Gamma_{lr}$) from 0.1 to 0.5 at 0.1 intervals. In the top panels with respect to BSR for (**a**) VOD = 0.1 and (**b**) VOD = 0.3, in the bottom panels with respect VOD for (**c**) BSR = 0.1 and (**d**) BSR = 0.4.

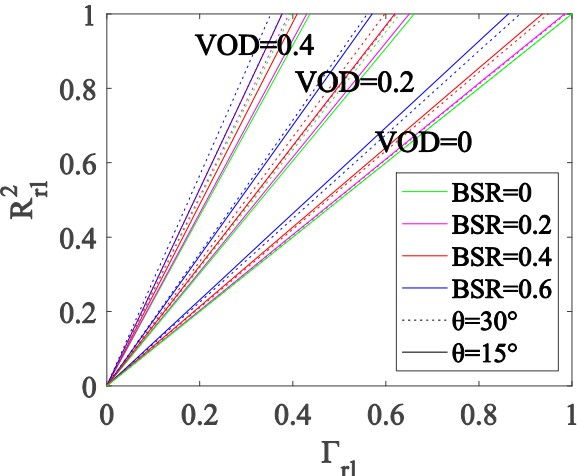

**Figure 8.** The sensitivity of the reflectivity correction ($R_{rl}^2$ vs $\Gamma_{rl}$) at a constant incidence angle ($\theta = 15°$ and $\theta = 30°$) for different BSR and VOD values.

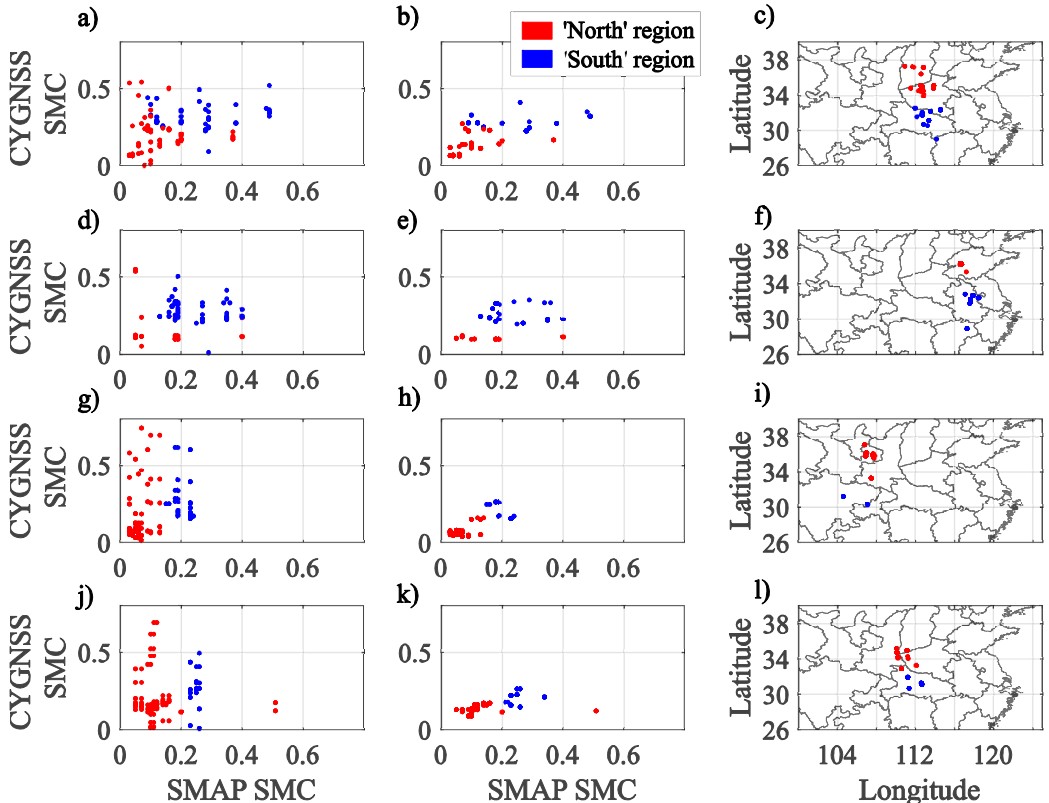

**Figure 9.** From top to bottom, SPs validation test (CYGNSS ∩ SMAP ∩ ICESat-2) using ICESat-2 roughness on 1 (panels **a**–**c**), 3 (panels **d**–**f**), 4 (panels **g**–**i**), and 6 (panels **j**–**l**) April 2019. In the left column (panels **a**,**d**,**g**,**j**), the scatter plot shows SMAP versus CYGNSS SMC estimates [cm$^3$·cm$^{-3}$] for the subset validation-test employing ICESat-2 roughness. In the middle column (panels **b**,**e**,**h**,**k**), the same subset as seen with SMAP roughness. In the right column (panels **c**,**f**,**i**,**l**), the corresponding locations for each subset. Northern regions (summer monsoon and dry winter, Cwa) are in blue and southern regions (without dry season, Cfa) are in red.

### 4.2. CYGNSS Derived SMC from ICESAT2 and SMAP/Sentinel1

We further compute CYGNSS SMC estimates employing BSR from ICESat-2 and compare the performance against CYGNSS SMC with SMAP BSR (CYGNSS ∩ ICESat-2 ∩ SMAP). The results are

shown in Figure 9, where the right panels show the resulting locations of each SP. The corresponding scatter plots for each case are shown in the left (CYGNSS SMC with ICESat-2 BSR) and the middle (CYGNSS SMC with SMAP BSR) panels (Figure 9). We divide the analysis for northern regions and southern regions. Higher SMC values are observed when employing ICESat-2 BSR. We can observe a clear difference range between SMAP and ICESat-2 BSR data, where the high sensitivity of ICESat-2 BSR ranges from 0 to 30 m, and that of SMAP BSR from 0 to 20 cm (see color bars in Figure 2). As a result, the corresponding correction to attenuation of vegetation increases as ICESat-2 BSR increases.

### 4.3. CYGNSS SMC Validation Using SMAP Data

Due to the above-mentioned lack of spatial coverage with the ICESat-2 BSR data, CYGNSS SMC has been definitely derived using BSR and VOD estimates from SMAP products. In this scheme, we first compute soil surface reflectivity values according to the suggested methodology. Concerning the incidence angle ($\theta$), we employ the model of [67], which allows computing soil permittivity by means of the linear $R_{hh}$ reflectivity, with best results within the limit of $\theta_i < 35°$, which verifies the observations made by [31]. In order to substantiate that the number of CYGNSS observations under these circumstances provides a sufficient number of SPs, Figure 10 shows the histogram for CYGNSS incidence angles on April 2019 limited to the study area. Assuming the combined geometry of the CYGNSS with the GNSS satellites, this estimate is assumed to be similar even on a daily basis. In this figure, the number of SPs with $\theta_i < 35°$ represents about 60% of the monthly population of observables. This percentage of SPs is significantly high in order to prove the suitability of the suggested SMC inversion process.

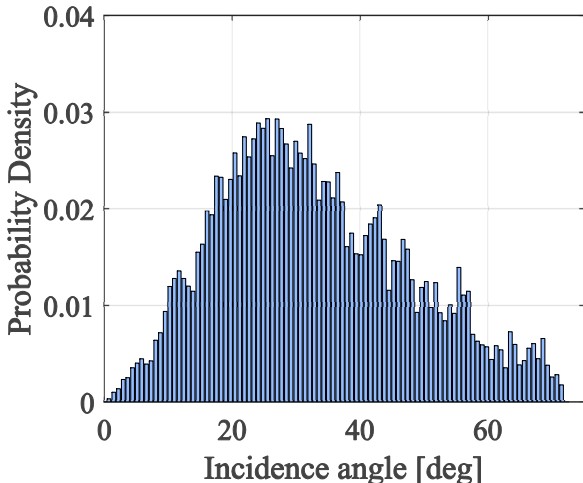

**Figure 10.** The empirical probability density for CYGNSS incidence angles during the whole month of April 2019 over the study area.

Once SMC is retrieved for the selected SPs, the final estimates are validated with SMAP SMC products, which are considered as reliable SMC reference [71]. For the CYGNSS data, the need for a bias value has already been pointed out in [42,43], and the authors in [37] provided an estimate for specific BSR and VOD values. Following these indications, we employ a reflectivity constant bias value of 0.15. With the finality to avoid unwanted biases in the validation process, since SMAP SMC products are derived using their own BSR and VOD (equivalent to VWC) products, we also employ these estimates to derive CYGNSS SMC. Moreover, we reduce the errors in the validation process by limiting VOD to ~0.5 for the southern regions ($\phi < $ ~32°N). This limit is reduced to about 0.3–0.4 depending on the rainfall probability. For the northern regions ($\phi > $ ~32°N), VOD is limited to about 0.3. Concerning the incidence angle ($\theta$), the model of [67] has provided the best results with a limit of $\theta < 35°$.

In few cases of high VOD, or under elevated rainfall probability, better results have been shown by limiting the incidence angle to lower values ($\theta < 15°$). These constraints agree with the above sensitivity analysis (Figure 7). Finally, the last filter involves looking for coincident locations between CYGNSS SP, SMAP, and/or ICESat-2 estimates in a range inferior of 500 m. Figure 11 shows the resulting set of 4568 validation SPs (CYGNSS ∩ SMAP) for April 2019 and the corresponding statistics in the linear regression model, showing strong linear evidence (Figure 11b) with zero p-value, small Root Mean Squared Error (RMSE = 0.05), and an R-squared = 0.6. Parameters of the regression model are shown in Table 1. The sparse distribution over the area of study is shown in Figure 11a.

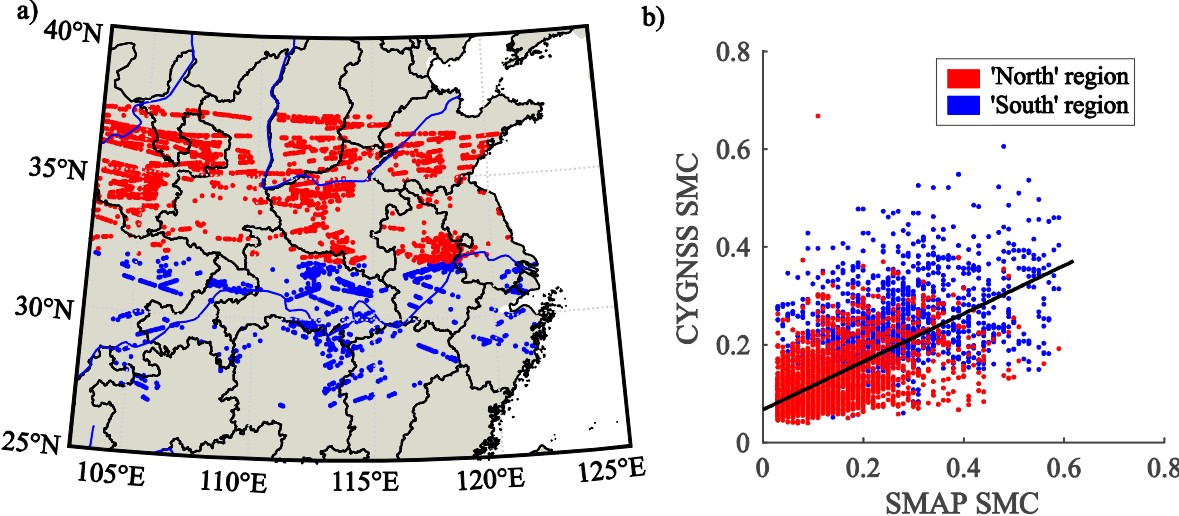

**Figure 11.** In (**a**) we see the location of the validation SPs (CYGNSS ∩ SMAP) for April 2019 (days shown in Figure 5). In (**b**), the scatter-plot shows the SMAP estimates versus the CYGNSS estimates of SMC [$cm^3 \cdot cm^{-3}$]. Parameters of the regression model are shown in Table 1. Northern regions (summer monsoon and dry winter, Cwa) are in blue and southern regions (without dry season, Cfa) are in red.

**Table 1.** Linear robust regression model of least absolute residual method: "$SMC_{SMAP} = \beta_0 + \beta_1 \cdot SMC_{CYGNSS}$". The corresponding scatter-plot is shown in Figure 11.

| Parameter | Value |
|---|---|
| Number of observations | 4568 |
| RMSE | 0.05 |
| R-squared | 0.6 |
| p-value | 0 |
| $\beta_0$ | 0.0669 ± 0.0290 |
| $\beta_1$ | 0.4916 ± 0.0135 |

Finally, Figure 12a shows an example of CYGNSS GNSS-R SMC estimates for 1 April 2019. We limit the data to SPs whose incidence angle is $\theta < 40°$ and employ SMAP VOD < 0.5 estimates from 1–12 April 2019. In Figure 12b the same day is shown for SMAP SMC estimates at 500-sample intervals. In this figure, the reader can see the higher temporal resolution and the capability of GNSS-R to estimate SMC, which, in contrast to SMAP SMC estimates, that need about a half a month to register the exact same area, these cover globally and at a continuous rate.

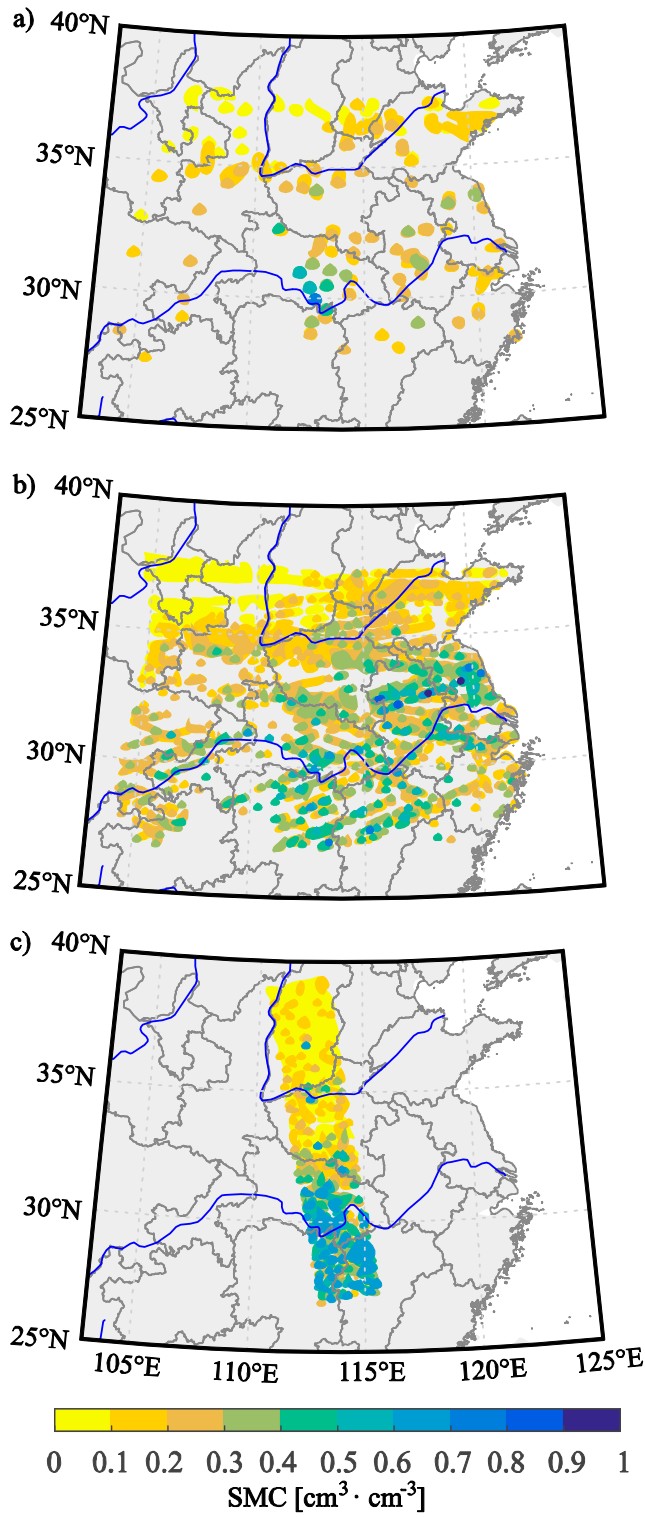

**Figure 12.** In (**a**) is shown SMC from CYGNSS for 1 April 2019, for $\theta < 40°$ and using SMAP VOD < 0.5 estimates from Figure 3 and ICESat-2 BSR estimates from Figure 2b. In (**b**) is shown SMC from CYGNSS for the same day, for $\theta < 40°$ and using SMAP VOD < 0.5 estimates from Figure 3 and SMAP BSR estimates from Figure 2a. In (**c**) is shown SMC from SMAP for the same day at 500-sample intervals.

## 5. Discussion

Recent advances on GNSS-R [19,21,34] have shown extraordinary capabilities to sense SMC [23–32] with high spatial and temporal coverage (Figure 12), low cost, and all-weather conditions, far to be

achieved by using conventional active/passive microwave instruments onboard satellites (e.g., SMAP, SMOS). However, the research community has been facing difficulties to define a reliable benchmarking GNSS-R SMC retrieval method (e.g., [27,37,41–44]) due to its inherent multidisciplinary character (GNSS technology, microwave remote sensing, measurement geometry and errors, spatial and temporal sampling, computation load, etc.). In response to this situation, this work presents a new methodology for converting reflectivity measurements from GNSS-R into Land Surface SMC estimates. This methodology differs from other studies where SMC was inappropriately related to GNSS-R reflectivity estimates and without physical basis, as for example, by deriving large time time-series of data, or doing a regression analysis to reliable SMC datasets (e.g., [19,27,37,41–44]). Instead, it has been proven that under some geometrical circumstances, i.e., for incidence angles below 30°, the inversion of the Fresnel reflectivity equations for linear polarizations is possible [31,45,47,51]. This can be achieved (makes sense) since most of CYGNSS' SPs are observed under these conditions (Figure 10).

According to the presented results, this strategy has some weaknesses that must be reviewed and solved in future work. First, improvements must be made at the level of the SMC estimation procedure. It is important to extend the SMC estimation to greater angles of incidence, not limited to 35°, and thus to increase the number of SPs. This implies reviewing methods such as those proposed in [72], in which an inversion method based on the two linear Fresnel coefficients for GNSS-R data is undertaken. The authors based their work on the coherent component of the signal without considering the incoherent BSR and VOD effects. However, these effects can be studied using bistatic radar models such as the one proposed by [63].

Regarding, the use of external data such as BSR and VOD, SMAP/Sentinel 1 L2 product appears to be suitable for agricultural areas. However, for forest or woodland areas additional efforts should be taken to supply more realistic estimates, e.g., ICESat-2 BSR. In this work, we derive CYGNSS SMC from different land cover types such as woodland, grassland and agricultural areas (Figure 4), and test two data sources for BSR estimates. On the other hand, BSR estimates from ICESat-2 are more sensitive to land surface microrelief, and the measurements are given along orbital paths, though generating large gaps between these tracks if no large time-series of data is employed. For instance, Figure 2b provides a one-month data set. However, since the BSR state might be considered stable along time, a sufficiently dense data set can be interpolated and employed for SMC retrieval. In this manuscript, our tests carried out with one month ICESat-2 BSR data have shown the high sensitivity in SMC retrieval to high BSR values. In fact, since SMAP SMC estimates employed in our validation test are derived from SMAP BSR data, the high sensitivity of ICESat-2 to land surface microrelief has provided a promising research gap for future studies on GNSS-R SMC retrieval. On the one hand, we employ SMAP SMC products to validate our new methodology, so it makes sense to employ SMAP BSR and VOD products for the validation test. The reason is that since SMAP SMC estimates are derived from its own BSR and VOD products, the resulting CYGNSS SMC estimates will not be affected by employing distinct BSR and VOD products.

Concerning the SMC validation strategies, additional reliable reference datasets are necessary. For example, in some particular conditions and geographical areas not frequently affected by clouds, reliable SMC estimates could be obtained by means of optical multispectral techniques, e.g., MODIS or Sentinel 3 products, which spatial and temporal resolutions can fulfill the GNSS-R SMC requirements. Moreover, since there are no satellite sensors that measure SMC directly, processing errors present in all these satellite-based SMC retrieval techniques requires a reliable calibration. In response to this problem, in situ measurements from reliable monitoring networks (e.g., National Soil Moisture Network, http://nationalsoilmoisture.com/) can be employed to calibrate the current satellite-based techniques, and more accurate and reliable results will be obtained. Other factors to take into account are those related to instrument calibration as pointed out in [37]. However, this question is beyond the scope of the here presented model design and implementation.

## 6. Conclusions

While further studies must focus on establishing a reliable benchmark to be used as an operational SMC retrieval method from GNSS-R data, as a consequence of our findings and investigations, the results achieved in this work are very encouraging. The conclusions derived from this study can be summarized as follows:

- A new method to retrieve SMC from GNSS-R is presented and validated, purely based on a bistatic radar physical modeling of the dielectric permittivity using Fresnel reflection coefficients and accounting the effects of BSR and VOD.
- This new approach is applied and tested with one month of CYGNSS GNSS-R data (April 2019 at the eastern region of China), in combination withICESat-2 and/or SMAP BSR and VOD products. The tests carried out with ICESat-2 BSR data have shown the high sensitivity in SMC retrieval to high BSR values, due to the high sensitivity of ICESat-2 to land surface microrelief.
- This CYGNSS SMC approach is validated with SMAP SMC products, and the statistical assessment provides an R-square of 0.6 (RMSE of 0.05 and zero p-value) for 4568 test points evaluated during April 2019 at the eastern region of China.

Future work will be conducted, but not restricted, to study sensitivity on SMC retrieval to different incidence angles, BSR, and VOD parameters, and assessing the reflectivity models. Additional SMC sources for validation should also be explored, focusing the effort to establish a reliable benchmark for SMC retrieval in terms of the above variables. Finally, CYGNSS calibration procedures and incorporation of incoherent scattering in the bistatic radar model must also be investigated.

**Author Contributions:** A.C. motivated the initial idea and carried out the computational programming, data analysis, experimental results, and draft preparation; I.M. materialized the initial concept though investigation of previous research, formulation of new theory and methodology, draft preparation, supervision of achieved results, funding support, and provided international academic exchange; S.J. provided supervision, mentorship, funding support and computational resources as well as undertook revision tasks. All authors have read and agreed to the published version of the manuscript.

**Funding:** This research was funded by Strategic Priority Research Program Project of the Chinese Academy of Sciences, grant number XDA23040100; the Jiangsu Province Distinguished Professor Project, grant number R2018T20, the Talent Start-Up Funding project of NUIST, grant number 1411041901010, the Startup Foundation for Introducing Talent of NUIST, grant number 2243141801036, and the R&D+I Program of the Universidad Politécnica de Madrid (ProgramaPropio UPM 2019).

**Acknowledgments:** Great appreciation is extended to ESA and NASA for providing the data access. Special thanks are given to the three reviewers for their constructive comments and helpful suggestions.

**Conflicts of Interest:** The authors declare no conflict of interest.

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
