# Peer review of "Soil Moisture Content from GNSS Reflectometry Using Dielectric Permittivity from Fresnel Reflection Coefficients"

_remotesensing, doi:10.3390/rs12010122_

Round 1
Reviewer 1 Report
This study proposes to estimate soil moisture from the Cyclone Global Navigation Satellite System-Reflectometry (CYGNSS-R) by reducing the effects of soil roughness and vegetation using ICESat-2 and Soil Moisture Active Passive (SMAP) products. The topic is not quite my field of expertise so that I am able to only provide some minor comments. Overall, the paper is well-written, the methodological description is fine, and the results are presented and discussed properly. Some English editing is required. Below, some minor comments:
In the proposed methodology, authors used the combination of GNSS-R, ICESat-2, and SMAP products to improve soil moisture estimation reducing the effects of soil roughness and vegetation water content and structure. Then, the final estimates were compared with SMAP Soil Moisture Content data for validation, considering that SMAP SMC data are reliable. Please, provide some references in this statement. Although SMAP SMC data are more reliable, CGNSS presents the advantage of having higher temporal resolution than that from the SMAP. This seems to be another important contribution of the study, thus, it should be highlighted in the Abstract and in the Conclusion. What is the meaning of “several hectares” in the text (Abstract and Introduction)? Please, be more specific. In Figure 6, please, identify the symbols in the title of the figure. I suggest the use of SMC rather than Mv since “SMC” has been used throughout the text. I believe the arrow linking ICESat-2 to BSR and the arrow linking SMAP to BSR/VOD/VWC are pointing towards the wrong direction. Figure 9. Is there any reason to split the data into North and South regions (for example, distinct rainfall event during the CYGNSS overpass or different conditions of soil roughness and vegetation biomass & structure)? Please, change the legend of this figure: “North region” first and “South region” after. Same for Figure 11. L510: please, change the word “indecently” (it is too strong!) to “inappropriately”.Author Response
Dear Reviewer,
Please see the attached document.
Sincerely,
Andres Calabia, Iñigo Molina, and Shuanggen Jin

Reviewer 2 Report
The manuscript is very well written, in a clear and scientific manner, and I have no hesitation in recommending it for publication.
However, there is one question I wish to ask. What is the justification for using GNSS-R for SMC when SMOS/SMAP are SM missions? The reason I am asking this is, neither SMOS nor SMAP 'observes' SM directly, and so the SM product would have some inherent 'errors' associated with the models being used. Yet when you use GNSS-R for deriving the SMC, wouldn't you be (in a way) adding more 'processing errors' to the derived SMC with the different data sets that are used as inputs (as each of these data would have some 'differences/errors' due to processing)?
Author Response
Dear Reviewer,
Please see the attached document.
Sincerely,
Andres Calabia, Iñigo Molina, and Shuanggen Jin

Reviewer 3 Report
The manuscript titled“ Soil Moisture Content from GNSS Reflectometry using Dielectric Permittivity from Fresnel Reflection Coefficients“ is well-written manuscript except for discussion and conclusion section. Paper could be very interesting for the reader of Remote Sensing. I have some issues with this manuscript and those should be clarified before the manuscript would be accepted.
Abstract
Most part of Abstract explains the background of the study it should be more concise. Abstract should contain introduction aim hypothesis aim result and conclusion Result in abstract should be in more detail. Sentence “ Nowadays, Global Navigation Satellite Systems-Reflectometry (GNSS-R) can retrieve SMC, while previous solutions of GNSS-R SMC retrieval were only based on direct comparisons, statistical regressions, or time-series analyses between GNSS-R observables and external SMC products.“ Is hard to read rewrite it.
Introduction
The introduction is written well. Below three points are my concern
Line no 56 should be “In this study“ instead “In this is study“ Line no 61 statement “ With the active devices, only soil dielectric properties can be assessed by means of measuring the scattering coefficient, usually denoted as the bistatic or back-scattering coefficient“ needs citation. Line no 61 should be “ by the means of“ instead “by means of“.
Discussion
Discussion is shallowly written it needs improvement with some more link with previous studies. Each section of result should be explained in the discussion with some support with the other studies.
Conclusion
The conclusion section is very lengthy written, conclusion should contain only the bullet points, not the story. Must be rewritten with some more unique information of this study.
Author Response

(The authors gave the same response as above.)

Round 2
Reviewer 3 Report
Please accept the manuscript in present form.